Journal of Machine Learning Research 1 (2021) 1-8          Submitted 7/18; Published 00/00

# SMILE: Sparse-Attention based Multiple Instance Contrastive Learning for Glioma Sub-Type Classification Using Pathological Images

*                                                                      *

*

*

*

**Editor:**

## Abstract

Gliomas are the most prevalent malignant brain tumor in adults and can be classified into four typical sub-types based on histological features. Histological diagnosis by pathologists via microscopic visual inspection of pathological slides has been the gold standard for glioma grading, especially hematoxylin and eosin (H&E) sections. However, due to spatial heterogeneity and complex tumor micro-environment, it is difficult and time-consuming for pathologists to differentiate glioma sub-types. In this paper, we propose a Sparse-attention based Multiple Instance contrastive LEarning (SMILE) method for glioma sub-type classification. First, we use contrastive learning to extract meaningful representations from pathological images. Second, we propose the sparse-attention multiple instance learning aggregator to get sparse instance representations in a bag for label prediction. We validate the proposed SMILE method using a glioma dataset from The Cancer Genome Atlas (TCGA). Experimental results show superior performance of our method over competing ones. Ablation study further demonstrates the effectiveness of our design of SMILE.

**Keywords:** Glioma classification, Sparse-attention, Multiple instance contrastive learning

## 1. Introduction

Gliomas are the most common primary brain tumor which comprise about 30% of all brain tumors and central nervous system tumours, and 80% of all malignant brain tumours. They are classified into four sub-types based on histological features by the World Health Organization (WHO) (Louis et al., 2016b): astrocytomas (A), oligodendrogliomas (O), glioblastomas (GBM), and oligoastrocytomas (OA) (see Fig. 1). Accurate classification of Gliomas using pathological images plays an essential role in therapy planning. Nowadays, this task is still mostly performed by pathologists via microscopic visual inspection of pathological slides, which requires considerable expertise and concentration, and it is also labor intensive and prone to bias. More important, despite well-established grading strategies, analyses from multiple pathologists on the same patient (especially those without significantly bifurcated appearance features) can easily yield inconsistency (van den Bent, 2010). Therefore, automated glioma sub-type classification is required to assist pathologists in efficient and effective diagnosis (Louis et al., 2016a).

Glioma sub-type classification is challenging in three aspects. First, whole-slide images (WSIs) (Mobadersany et al., 2018) have a huge size, ranging from several 1 000 to 10 000 pixels along each direction. The pixel-level dense annotations of WSIs are very limited, if available, which makes this classification problem a weakly supervised one. Second, different from other kinds of tumor slides, glioma diagnostic slides are occupied almost entirely by tumorous cells of various morphologies. When dividing a high grade WSI into small patches, some patches may contain the visual features from lower grade tumors. The spatial heterogeneity makes the weakly supervised task more difficult. Third, the tumor micro-environment is complex.

In recent years, convolutional neural networks (CNNs) have shattered performance benchmarks on image classification tasks (Mobadersany et al., 2018) and show new opportunities in histopathological glioma sub-type classification (Janowczyk and Madabhushi, 2016). Ertosun and Rubin (2015) applied CNNs to binary classification of glioblastoma and low-grade glioma (LGG). Jin et al. (2021) proposed a squeeze-and-excitation block DenseNet (SD-Net) to classify five sub-types of gliomas. They selected 300 patches per WSI, which limits the performance of this method. To address the issue of spatial heterogeneity, multiple instance learning (MIL) has been successfully applied to computational pathology for tasks such as tumor classification (Campanella et al., 2019; Chikontwe et al., 2020; Hou et al., 2016; Li et al., 2021). Campanella et al. (2019) proposed a MIL classifier trained on the large weakly-labeled WSI datasets and achieved better performance than the fully-supervised classifier trained on small pixel-annotated lab datasets. Ilse et al. (2018) proposed an attention based multiple intance learning (ABMIL), which only consider self-correlation of instances. Actually, pathologists often consider both he contextual information around a single area and the correlation information between different areas when making a diagnosis decision. Vaswani et al. (2017) proposed transformer based on multi-head self-attention for language tasks, which uses relation matrix to calculate instance correlation. Dosovitskiy et al. (2020) proposed an version transformer, which makes it possible to use transformer in vision tasks. Note that, self-attention based transformer is much desirable to consider the correlation between different instances in MIL aggregator.

Recently, contrastive learning has demonstrated success in learning visual representations without labels (Chen et al., 2020; He et al., 2020; Grill et al., 2020). It is very suitable to use a contrastive strategy to learn feature representations of pathological patches. This strategy can divide pathological patches into different classes without labels. To use unlabeled pathological images for model training, we advocate to explore contrastive learning for glioma sub-type classification. In this paper, we proposed a novel approach, called Sparse-attention based Multiple Instance contrastive LEarning (SMILE), for glioma sub-type classification using pathological images. First, we use a contrastive training strategy to pretrain a patch-level feature extractor so that we can obtain discriminative patch-level feature representations. Second, we propose to use sparse-attention multiple instance learning and get better classification by using a sparse-attention block. We validate our proposed method using the TCGA glioma dataset and the accuracy of sub-classification is 0.8857.

The contributions of this work are summarized as follows: (1) first use contrastive learning strategy to learn feature representations on glioma sub-type classification, and (2) propose a sparse-attention block for multiple instance feature aggregation.

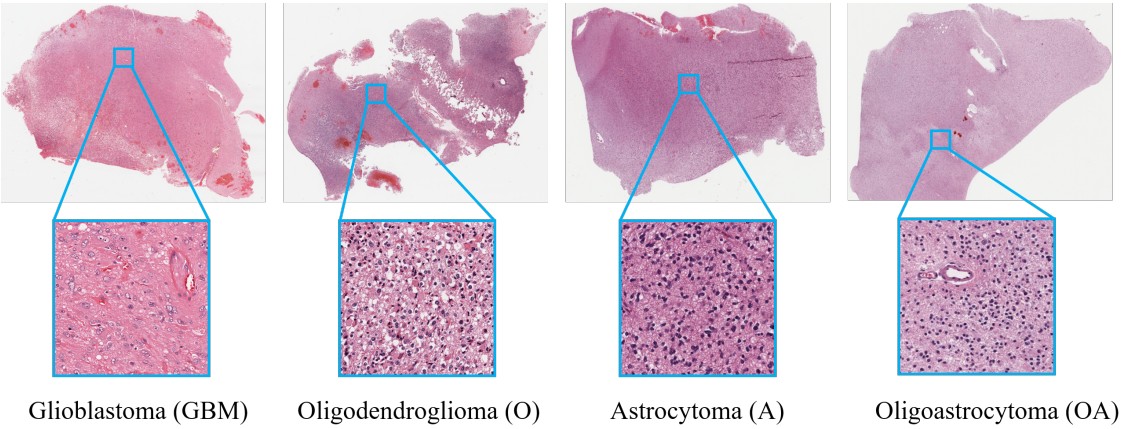

| Glioblastoma (GBM) | Oligodendroglioma (O) | Astrocytoma (A) | Oligoastrocytoma (OA) |

Figure 1: Examples of four glioma sub-types. Each legend has two parts: Left part is at 0.3× magnification, and right part is at 10× magnification.

## 2. Methods

In this section, we introduce our proposed Sparse-attention based Multiple Instance contrastive LEarning (SMILE) framework for glioma sub-type classification. The proposed SMILE framework consists of two main components: (1) a contrastive learning strategy to train a powerful feature extractor, and (2) a sparse-attention block for meaningful multiple instance feature aggregation.

### 2.1 Contrastive Learning for Better Feature Representation

Contrastive learning is a popular research topic recently, since it enables learning robust feature representations without manual labels. In particular, two random transformations are applied to one training image for obtaining a pair of augmented images. Each transformed image from the pair of images is then fed into an encoder, and is finally projected to latent feature representations. We constrain the latent features from the same image to be close while those from different images to be far-away. Inspired by this concept, we try to pretrain a powerful patch-level feature extractor to learn patch-level discriminative feature representations. However, almost all glioma image patches from the slide have tumor tissue, so these patches are positive samples. It is quite different from the typical contrastive learning scenario which requires both positive and negative samples. As a consequence, we build our own contrastive learning module (as shown in Fig. 2) by borrowing the idea from a contrastive learning paradigm that does not require negative samples in a batch(Grill et al., 2020). We will introduce the building blocks of this module one by one below.

**A stochastic data augmentation module** transforms any given data sample randomly and results in two correlated views for the same sample, denoted $x_i$ and $x_j$, which we consider as an input pair. We sequentially apply simple augmentations: random cropping, followed by resizing back to the original size, random color distortion, random rotation, random flip, and random Gaussian blurring.

**A feature encoder** $E$ extracts representation vectors from augmented data samples. Our framework is flexible to various network architectures without much constraint. We opt for ResNet-50 (He et al., 2016) as the encoder backbone, which contains a convolutional (Conv) layer, four stages of residual blocks, an average pooling layer, and a fully connected (FC) layer, followed by softmax. We remove the FC layer and softmax. ResNet-50 is used to obtain feature embedding $h_i$, where $h_i \in R^{1 \times 2048}$ is the output after the average pooling layer.

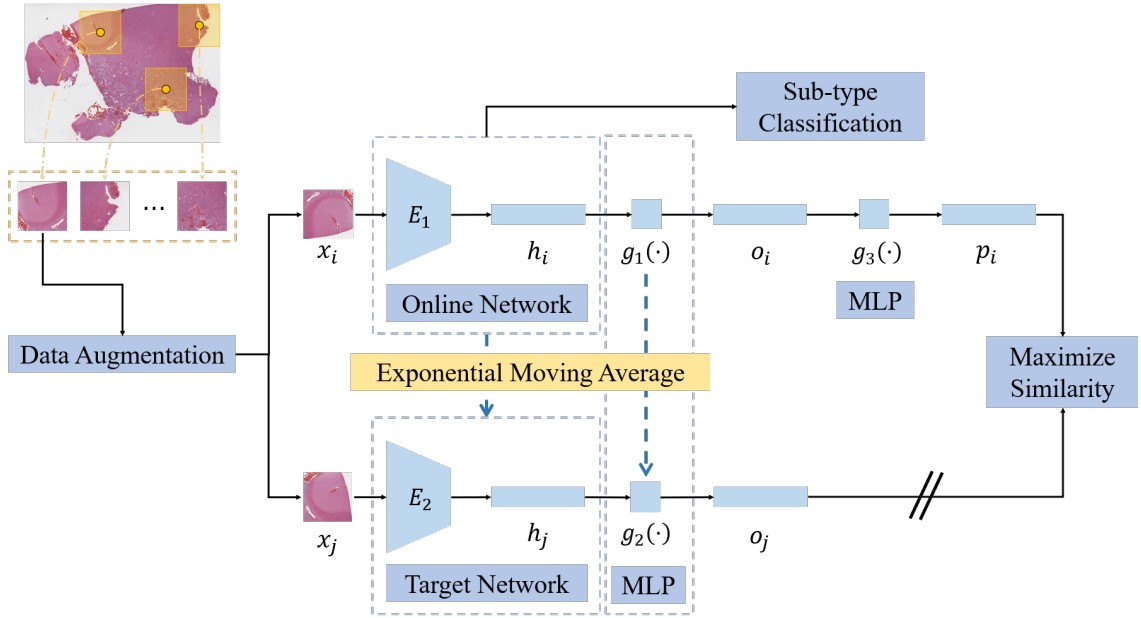

Figure 2: The workflow of contrastive learning for pathological images. The target network weights are updated by an exponential moving average of the corresponding online network weights. We take one patch and randomly transform to get a pair of patches, i.e., $x_i$ and $x_j$. After the feature extractors $E_1$ and $E_2$, we get representations $h_i$ and $h_j$. $p_i$ is obtained by using two MLP operations, $g_1(\cdot)$ and $g_3(\cdot)$. $o_j$ is obtained using one MLP operation $g_2(\cdot)$. The symbol $//$ means stop-gradient. The pathological patches selected from the WSI are used for contrastive training.

**A multi-layer perceptron (MLP)** maps representations to the space where contrastive loss is applied. The MLP $g(\cdot)$ consists of a $1 \times 1$ Conv layer with the output size of 4096, followed by batch normalization (BN), rectified linear units (ReLU), and a $1 \times 1$ Conv layer with the output dimension of 256.

**A contrastive loss function** defines for a contrastive prediction task. Given an image $x$ and a pair of transformed examples $x_i$ and $x_j$, the contrastive prediction task aims to maximize the similarity of a given pair, i.e., $x_i$ and $x_j$. From the augmented $x_i$, the online network outputs a representation $p_i$, and from the augmented $x_j$, the target network outputs $o_j$. We then use $l_2$ normalization on both $p_i$ and $o_j$ to get $\hat{p}_i$ and $\hat{o}_j$. The loss function for

an augmented pair is defined as

$$Loss_{i,j} = \parallel \hat{p}_i - \hat{o}_j \parallel_2^2 \tag{1}$$

**An exponential moving average** defines the weights update. The target network just provides the ground truth of the online network, and does not perform gradient propagation in each training step. The weights of target network $\mu$ is updated by an exponential moving average of the corresponding online network weights $\theta$. There are given by:

$$\mu = \omega\mu + (1 - \omega)\theta \tag{2}$$

where $\omega$ denotes the target decay rate.

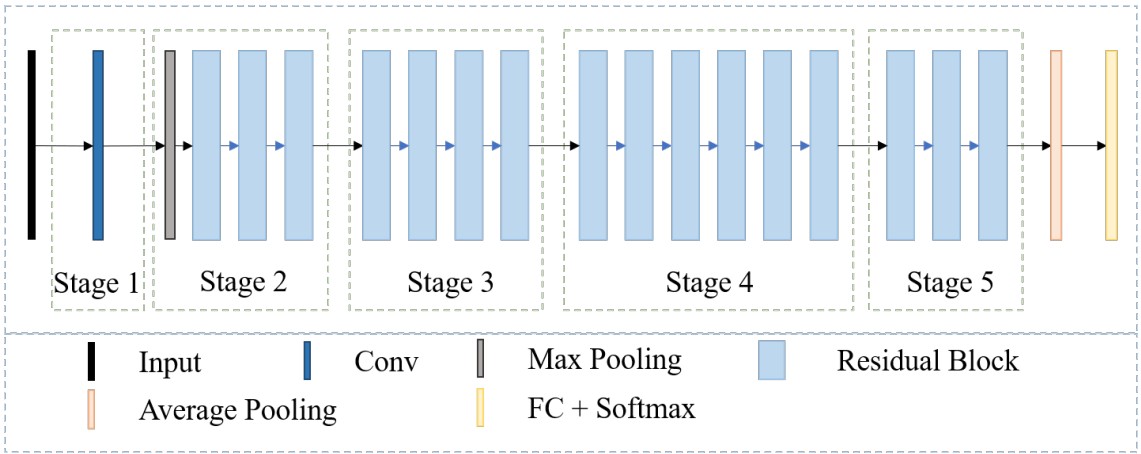

Figure 3: The workflow of ResNet-50. It contains a convolutional (Conv) layer, four stages of residual blocks, an average pooling layer, and a fully connected (FC) layer, followed by softmax.

## 2.2 Sparse-Attention based Multiple Instance Learning

Different from most previous methods which either learn an instance classifier or a bag classifier, our proposed Sparse-Attention Module (SAM) based multiple instance learning jointly learns the instance classifier, the bag classifier, and the embeddings in one architecture. Our method includes three parts: instance-level embedding and classification, instance embedding aggregator, and slide-level classification.

Let $A = \{a_1, a_2, \cdots, a_m\}$ be a bag of instances where $a_i$ is the ith instance. Let $H = \{h_1, h_2, \cdots, h_m\}$ be feature embeddings of instances where $h_i = E(a_i)$ is obtained by the feature extractor $E$. The first part is an instance-level classifier operating on each of the instances. The output $B = \{b_1, b_2, \cdots, b_m\}$ is obtained by $b_i = \mathbf{W}h_i$, where $\mathbf{W}$ is a weight vector.

Then we select the top-n instance embeddings $\hat{\mathbf{H}} = \{\hat{h_1}, \hat{h_2}, \cdots, \hat{h_n}\}$ as the following transformer inputs.

In the second part, we aggregate the instance embeddings into a bag embedding which is further scored by a transformer classifier, as shown in Fig. 4. We will introduce the transformer classifier blocks below.

**A multihead self-attention module (MSA)** contains several parallel self-attention (SA) layers. Each self-attention layer can be described as mapping a query and a set of key-value pairs to an output, where the query $\mathbf{Q}$, key $\mathbf{K}$, value $\mathbf{V}$ are given by:

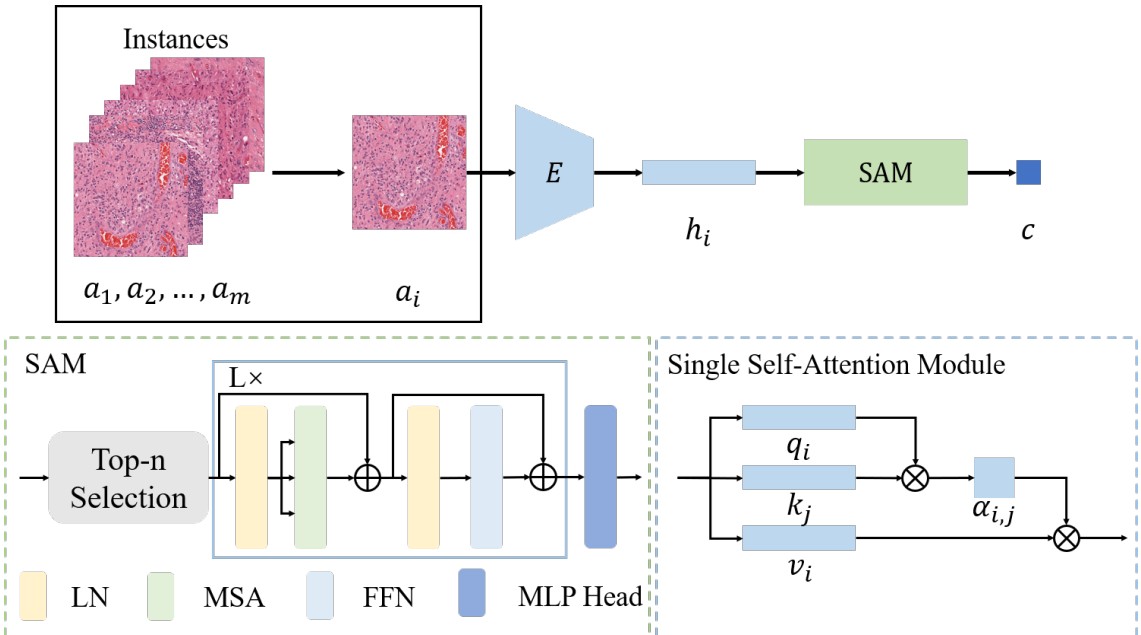

Figure 4: The workflow of SAM module. A batch of instances from one bag will be input in an extractor that is trained by contrastive learning and get the feature embeddings. We use the SAM module to select the top-n instance feature embeddings, and then input the embedding into the transfomer module, which contains of alternating layers of LN,MSA and FFN blocks. We use a MLP head to predict the final output. The single SA module is shown in bottom-right corner.

$$\mathbf{Q} = \mathbf{W}^q \hat{\mathbf{H}}, \mathbf{K} = \mathbf{W}^k \hat{\mathbf{H}}, \mathbf{V} = \mathbf{W}^v \hat{\mathbf{H}}, \tag{3}$$

where $\mathbf{W}^q, \mathbf{W}^k, \mathbf{W}^v$ are weight vectors. We compute the single self-attention function of output as:

$$\text{Attention}(\mathbf{Q}, \mathbf{K}, \mathbf{V}) = \text{softmax}(\frac{\mathbf{Q}\mathbf{K}^T}{\sqrt{d}})\mathbf{V}. \tag{4}$$

Multihead s1lf-attention fuction is computed by:

$$\text{MultiheadAttention}(\mathbf{Q}, \mathbf{K}, \mathbf{V}) = \text{Concat}(\text{attention}_1, \text{attention}_2, \cdots, \text{attention}_m)\mathbf{W}^0. \tag{5}$$

The selected instance embeddings $\hat{h}_i$ is transformed with position encoding. The input of transformer $\mathbf{z}_0$ is defined as:

$$\mathbf{z}_0 = [h_{\text{class}}; \hat{h}_1; \hat{h}_2; \cdots; \hat{h}_n] + \mathbf{H}_{pos}, \tag{6}$$

where $h_c lass$ denotes the class of WSI and $\mathbf{H}_{pos}$ denotes position information of each instance.

The output features of MSA is computed by:

$$\mathbf{z}_\ell^{'} = \text{MSA}(\text{LN}(\mathbf{z}_{\ell-1})) + \mathbf{z}_{\ell-1}, \ell = 1 \cdots L \tag{7}$$

**A feed forward Network (FFN)** consists of a $1 \times 1$ Conv layer, followed by a Gaussian Error Linerar Unit (GELU), and a $1 \times 1$ Conv layer.

$$\mathbf{z}_\ell = \text{FFN}(\text{LN}(\mathbf{z}_{\ell-1}^{'})) + \mathbf{z}_\ell^{'}, \ell = 1 \cdots L \tag{8}$$

The MSA and FFN modules are repeated $L$ times.

**A multi-layer perceptron (MLP) head** The MLP head consists of a LN and a $1 \times 1$ Conv layer with the output dimension of class number $c$.

$$\mathbf{c} = \text{MLP}(\text{LN}(\mathbf{z}_L)). \tag{9}$$

## 3. Experiments and Results

### 3.1 Datasets

To validate our proposed multimodal paradigm for integrating histological and genomic features, we collect glioma data from the TCGA (Tomczak et al., 2015), a cancer data consortium that contains paired high-throughput genome analysis and diagnostic whole-slide images with ground-truth histologic grade labels. Subtype cases of A, O and GBM are in the merged TCGA-GBM and TCGA-LGG (TCGA-GBMLGG) project. There are 769 cases in this dataset, which contains 141 A cases, 209 O cases, 350 GBM cases, 36 OA cases, and 33 no-reported cases. Glioma whole slide images were cropped at $20\times$ objective magnification using OpenSlide [23]. High-power fields (HPFs) at $256\times256$ pixels were sampled from these regions and used for training and testing. We perform three-sub-type classification task, i.e., A, O, GBM classification. We divide the dataset into two parts: 560 slides (80%) for training, and 140 slides (20%) for testing. And the same proportion of each sub-type is set both in the training set and testing set, as shown in Table 1. There are 113 A slides, 167 O slides, and 280 GBM slides in each training fold. And there are 28 A slides, 42 O slides, and 70 GBM slides in each testing fold. We use 4-fold cross-validation.

| Class | Training Fold | Testing Fold |
|-------|---------------|--------------|
| A | 113 | 28 |
| O | 167 | 42 |
| GBM | 280 | 70 |
| Total | 560 | 140 |

Table 1: TCGA dataset of training fold and testing fold.

### 3.2 Implementation details

For pretraining with contrastive learning, we use Adam (Kingma and Ba, 2014) optimizer with a constant learning rate of 0.0001 to update the weights of the encoder model during the training. The batch size for training is 256, and the epoch is set to 100. The feature extractor is trained on a workstation with two NVIDIA GTX 1080Ti GPUs and 64GB Memory. The SAM is trained by using Adam with a constant learning rate of 0.0001. The number of self-attention head is set to 6 and the number of transformer block is set to 4. The batch size for training this part is 256, and the epoch is set to 80. The model is implemented on PyTorch.

| Method | Accuracy | F1 |
|---|---|---|
| ResNet-50+MIL | 0.7714 | 0.8624 |
| ResNet-50+MIL+Contrast | 0.7929 | 0.8710 |
| ResNet-50+ABMIL | 0.8214 | 0.8917 |
| ResNet-50+ABMIL+Contrast | 0.8357 | 0.8971 |
| ResNet-50+SAM | 0.8500 | 0.9074 |
| ResNet-50+SAM+Contrast (SMILE) | **0.8857** | **0.9266** |

Table 2: Performance of contrastive learning strategy, evaluated on the average of testing folds.

**Impact of contrastive learning strategy.** To investigate the impact of contrastive learning, we directly conduct experiments to compare the models with or without contrastive learning. The accuracy of ResNet-50 with MIL and contrastive learning strategy is 0.7929 and the F1 score is 0.8710. It improves the accuracy by 2.15% and the F1 score by 0.86% over the ResNet-50 with MIL. The ABMIL with contrastive learning strategy improves the accuracy by 1.43% and the F1 score by 0.54% over the single ABMIL. The SMILE improves the accuracy by 3.57% and the F1 score by 1.92% over the SAM module.

**Impact of SAM.** As shown in Table 2, the proposed SAM module improves the accuracy by 7.86% over the MIL and 2.86% over the ABMIL. Besides, the SAM module improves the F1 score by 4.5% over the MIL and 1.57% over the ABMIL, which demonstrates the effectiveness of the proposed SAM.

Generally, each proposed module in our method is proved to be effective.

## 4. Conclusion

In this paper, we proposed a SMILE framework for glioma sub-type classification. In particular, we use contrastive learning to pretrain a glioma sub-type oriented feature extractor, so that we can learn meaningful patch-level representations. Also, the proposed sparse-attention based multiple instance learning framework brings high performance gain for pathological image classification. Experimental results on clinical data show the effectiveness of our proposed method. In the future work, we will apply our proposed model to multiple caner types for further evaluation.

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
