# OpenReview forum: "SMILE: Sparse-Attention based Multiple Instance Contrastive Learning for Glioma Sub-type Classification Using Pathological Image"
_MICCAI.org/2021/Workshop/COMPAY — COMPAY 2021_

### Official Review · Reviewer_Qsum · 2021-08-16
**Contrastive learning in glioma classification**

**Rating:** 8
**Confidence:** 4

**Review:**

In this study, the authors propose a new algorithm to classify histopathology images of glioma in different subtypes. They obtain a ground truth on the level of whole slides, generate patches from these slides and train supervised classifiers to distinguish three classes of tissue. Subsequently, they aggregate predictions on the level of patients and assess the accuracy. Comparing their workflow to previously published methods, they demonstrate a superiority in terms of Accuracy and F1 score.

The new algorithm is based on contrastive learning which results in a better feature representation in neural networks which in turn enables a better classification performance. Contrastive learning is still fairly new in the field of computational pathology and this work is innovative. Also, they perform a credible benchmark comparison to five other methods, quantitatively demonstrating the superiority of their method. The authors use only the publicly available TCGA archive which is known to suffer from biases and batch effects, but is the de-facto standard benchmark for such technical studies.

The article could benefit from a more extensive validation in more patients cohorts as well as a more detailed reporting of statistical outcomes, including a ROC analysis. Yet, it is a technically novel and clinically interesting approach and scientifically sound so I recommend it for acceptance.

---

### Official Review · Reviewer_BByk · 2021-08-25
**Sparse-attention based multiple instance contrastive learning**

**Rating:** 6
**Confidence:** 5

**Review:**

To overcome the challenges of limited annotations, spatial heterogeneity and complex micro-environment in pathological image classification, the authors proposed SMILE algorithm, which is interesting for underlying application. The major contribution of this research is the integration of attention-based transformer, contrastive learning and multiple instance learning for gliomas histopathological image classification, and the results also show good performance.

Please clarify these doubts and consider following suggestions.

1.	Unbalanced data has always been the challenge for pathological image processing. However, this article ignores the normal categories of the Gliomas classification task and may neglect the negative samples in the design of the contrastive strategy, which requires justification for clinical diagnosis.

2.	Although the proposed method is interesting, this paper lacks rigorous experimental results and comparison with other SOTA methods, especially the ablation study of each module. In addition, some recent work should also be discussed in the experimental section, such as,
Shao Z, Bian H, Chen Y, et al. TransMIL: Transformer based Correlated Multiple Instance Learning for Whole Slide Image Classication[J]. arXiv preprint arXiv:2106.00908, 2021.
Li B, Li Y, Eliceiri K W. Dual-stream multiple instance learning network for whole slide image classification with self-supervised contrastive learning[C]//Proceedings of the IEEE/CVF Conference on Computer Vision and Pattern Recognition. 2021: 14318-14328.
Wang, Shujun, et al. RMDL: Recalibrated multi-instance deep learning for whole slide gastric image classification. Medical image analysis 58 (2019): 101549.

3.	The author claims “jointly learns the instance classifier, the bag classifier, and the embeddings in one architecture”, is this architecture an end-to-end model? Please further explain the training strategy of feature encoder and SAM. Specifically, how to achieve sparse representation?

4.	Looking at Table 2, it appears questionable to me why choose ResNet-50 as the encoder backbone instead of DenseNet mentioned in the introduction or other backbones, they should be introduced into experiments for comparison or to improve performance. In addition, what is the purpose of Figure 3? Does the author have a structural design to improve ResNet-50?

5.	Hyperparameter settings need to be rigorously discussed, for example, the number of self-attention head and transformer block.

6.	The attention module has been cited with several names in this paper, please explain the difference between sparse-attention and self-attention.

7.	Please elaborate on the third challenge “micro-environment is complex” mentioned in page 2. How was it solved by the proposed SMILE?

8.	How to understand “To validate our proposed multimodal paradigm for integrating histological and genomic features” in section 3.1? how to define “multimodal”? Does this article involve genetic data?

9.	Please check all grammar and spelling carefully.
Typo:
Page 2. “multiple intance learning (ABMIL)” should be “multiple instance learning (ABMIL)”
Page 6. “input the embedding into the transfomer module” should be “input the embedding into the transformer module”
Page 6. “Multihead s1lf-attention fuction is computed by” should be “Multihead self-attention function is computed by”

---

### Official Review · Reviewer_dTqc · 2021-08-25
**A method based on contrastive learning and sparse attention for the classification of glioma sub-types.**

**Rating:** 4
**Confidence:** 4

**Review:**

This paper proposes a method based on contrastive learning (to pre-train a feature extractor) and sparse attention multiple instance learning for the classification of glioma sub-types.

The novelty is limited but the task is relevant and the method is well suited for the task.

The paper should be better written (various typos and some grammar mistakes).

The method description lacks some motivation for the different blocks and has many technical flaws including:
- notations not defined (e.g d in eq (4), LN(.), W^0, attention_1...attention_m),
- eq. (5) not mathematical,
- Number of class c and and output c bold
-The method description also lacks some details: e.g. range of the transforms in the data augmentation module, the different dimensions of the vectors, layers etc.

The data splits are unclear. A fixed training and test splits are given, then a 4-fold cross-validation is mentioned without saying what for. In table 2, an average of testing folds is then mentioned. (also if it were an average, the standard deviation would be required)

Other:
Fig. 3 is useless and is not referenced in the text.
The two parts of the method are repeated too often without giving any further insight (end of 1-Introduction and beginning of 2-Methods)
You mention FC layers in the CNN and then 1x1 conv in an MLP. It is also an FC layer.
Page 7, why calling it a feed forward network? CNNs and MLPs are both FFN.

---

### Decision · Program_Chairs · 2021-08-25

Accept